# Outmigration Drives Cropland Decline and Woodland Increase in Rural Regions of Southwest China

**Yi Yu** [1,*] , **Tingbao Xu** [1] **and Tao Wang** [2]

1 Fenner School of Environment & Society, Australian National University, Acton, ACT 2601, Australia; tingbao.xu@anu.edu.au

2 College of Resources and Environment, Southwest University, Chongqing 400715, China; xndx2018@email.swu.edu.cn

\* Correspondence: u6726739@anu.edu.au; Tel.: +61-0449193587

**Abstract:** Rapid urbanisation in China has led to massive outmigration in rural regions, which has changed the regional labour force structure and can have various profound impacts as a result. This research used a case study in Southwest China to investigate how regional land use patterns have been changed in the context of rural outmigration and assessed the resulting dynamics on local ecological environment. The key findings include: (1) The local land conversion process was mainly characterised by the conversion of farmland (−18.3%) to residential area (+268.3%) and woodland (+55.6%) during 2000–2018; (2) about 83.7% of area showed a statistically significant increase in the normalised difference vegetation index (NDVI), which was not due to human interference factors (e.g., afforestation). Correlation analyses showed that depopulation (R = −0.514, $p < 0.01$) and local mild temperature (R = 0.505, $p < 0.01$) could be the main contributors. Only 2.5% of the area had decreased NDVI and this was directly caused by human activities (e.g., urban area expansion). These results implied that vegetation improvement can occur in the context of depopulation and farmland reduction, which did not significantly threaten the local agricultural sector. It then could be a good choice to allow those high-slope and biophysically poor farmlands to undergo forest succession rather than cultivation. Farmers in Southwest China should make a full use of the existing low-slope arable land to curb the declining trend of local farmland amount, in order to meet the future challenges brought by urbanisation. Enhanced agricultural infrastructure, mechanised farming and guide from local government can help achieve this goal. This study provided new insights and more realistic scenarios for rural development in Southwest China. The research findings are expected to provide a better understanding to enable sustainable land use management in Southwest China.

**Keywords:** land use and land cover change; urbanisation; rural outmigration; NDVI; Mann–Kendall test; linear regression; Southwest China

## 1. Introduction

Urbanisation can lead to rural outmigration, changing the size and structure of the rural labour force and hence will influence regional landscape patterns. Rural landscape structures are always affected by the dynamics of the agricultural sector in general and of the farming systems in particular [1], where the intensity of farmers' activities play a vital role. The intensity of agricultural activities could be weakened within the context of rural outmigration, posing risks to the amount of arable land and food security in rural areas. This phenomenon has been widely observed in various developing countries worldwide [2–4]. Among them, China's agricultural output needs to feed 20% of the world's population with only 7% of the arable land worldwide [5]. Its scarce arable land per capita, together with its huge population base (1.393 billion in 2018; [6]) and the small-scale household-based

agriculture, make China's rural problems somewhat unique. As a traditional agricultural country, China has always attached great importance to its agriculture, which has a profound influence on rural farmers' perception and activities. For instance, some studies claimed that Chinese farmers have always considered the farmland as their 'lifeblood' [7,8], and as a result, farmers always made considerable efforts to obtain high agricultural returns from the land. However, the relationship between farmland and farmers is experiencing a fundamental change, which can be attributed to China's current ultra-rapid urbanisation process [8]. Large numbers of farmers are choosing to leave their land to work in the cities, which will inevitably have a series of impacts on the rural landscape, agriculture, and the ecological environment. Liu and Gu [9] pointed that provinces in Central and Western China were often the main sources of migration since 1995; while the destinations of migration were mainly concentrated in the developed coastal provinces, such as the Yangtze River Delta and the Pearl River Delta. In 2015, 45.9% of rural migrants were interprovincial migrants, where those who chose to work in prefecture-level cities accounted for 80% [10]. The long spatial distance makes it difficult for these migrants to take care of their land once they moved out. According to the Chinese National Bureau of Statistics [11], the total number of migrant workers who had left their villages and hometowns, for greater than 6 months, reached 171.85 million in 2017. Nearly 45% of the rural-to-urban migrants in 2010 were born after the 1980s [12]. These peasant migrants, mainly young and fit men, have been attracted by the stable and higher paid employment found in urban areas [13,14] and have chosen to transfer their cropland to others or have abandoned their farms [15]. Once they moved out, a significant part of them were reluctant to return to the rural areas [12].

Rural outmigration in China is also often chained: firstly, a young and educated member of a rural family moves to urban areas to find off-agricultural employment opportunities, followed by the remaining adults in the family, and finally the entire family moves out [12]. In China, family migration is the mainstream of the rural-to-urban migration, occupying about 66% of the total migrants in 2010 [16]. This means that there exists a decreasing trend in both the rural labour force and the number of children growing up in the rural areas. As a price of urbanisation, China's rural population proportion has decreased from 82.10% in 1978 to 42.04% in 2017 [17]. The massive rural outmigration of young and skilled labour has given rise to brain drain, aging population, and stagnation in rural areas [18]. The size of labour force engaged in the primary sector has been shrinking rapidly from 355.75 million in 2000 to 202.57 million in 2018 [19]. China's Ministry of Agriculture and Rural Affairs pointed out that the average age of current front-line agricultural labour is nearly 53, where farmers older than 60 accounted for more than 25% [20]. As the agricultural mechanisation level is still low in many China's rural regions, the countryside will face the question that who will cultivate the farmlands. Furthermore, within the remaining rural population, a large proportion are the migrant workers' parents and children with a weaker labour capacity, also known as the left-behind people [21]. In 2015, the left-behind population in China's rural areas included 58 million children, 47 million women, and 45 million elderly people [14]. Most struggle to manage or sustain agriculture by themselves. As a consequence, in those areas where there have been large population movements (in particular young labour), smallholder agricultural practices and the villages' self-managing capacity could have been affected [14,21]. This has substantially weakened and modified agricultural activities and transformed land use patterns in rural regions. The traditional smallholding and labour-intensive agricultural module, particularly in South China, where fragmented farmlands are distributed on hilly terrains, makes China's rural agricultural activities vulnerable to rural depopulation.

Changes to population size and structure in rural regions will not only have social and economic impacts but will also result in changes to the natural environment and ecosystem of these regions. Farmers' behaviours are the key factors affecting landscape change in rural areas [22]. Changes in anthropogenic disturbance can inevitably lead to land use and cover change (LUCC) [23]. There have been numerous LUCC studies worldwide (e.g., Europe, India, Mexico, Nepal, Russia, and South Africa) that explored the rural outmigration phenomenon and its consequences [24–27]. In Nepal, Khanal and Watanabe found that rural outmigration and other social–economic factors (e.g., reduced crop yield,

regional policies) have resulted in farmland abandonment [24]. This abandonment of rural land posed challenges to food security, led to exacerbated poverty among rural households, and resulted in geomorphic damage within Nepal villages [24]. In the context of the collapse of the Soviet Union, the reforms of the agricultural sector and rural depopulation triggered widespread abandonment of farmland in Central and Eastern Europe [25]. Alcantara et al. found that 52.5 million ha farmland had been abandoned between 1991 and 2005, accounting for one-fifth of the total farmland in this region [25]. The rural outmigration in Latin America has transformed the distribution pattern of forests and the livelihoods in local communities [26]. Especially in Mexico, the rate of both rural outmigration and remittance economies is high, which has been discouraging its historical smallholder agriculture but created opportunities for vegetation recovery [27]. García-Barrios et al. found that this phenomenon could help curtail and even revert the decreasing trend of tropical forests, mainly through vegetation regeneration on those abandoned fields [27]. The findings of these region-specific studies can be instructive for the global rural outmigration research. However, most of them were based on specific historical contexts and typical regional characteristics. There are still many unanswered questions regarding the outmigration-related landscape transitions in the rural regions of Eastern Asia, especially in China, a historically agricultural country but currently experiencing unprecedented urbanisation progress. Issues here related to rural outmigration and corresponding consequences on LUCC and ecological environment have not been well integrated, with only a few studies associated with rural regions [2,28]. De Sherbinin et al. emphasised that it is urgent to conduct such research in Asia to understand how rural resources have been influenced by outmigration [29]. There are also a lack of studies in this field about how Asian rural land use and land cover (LULC) have been affected by such outmigration processes [29]. A better understanding of the impacts of emigration on rural regions has a particular significance to a country like China where maintaining a robust agricultural sector is required to feed the large population and to provide a foundation for the potential restoration of deteriorated ecological environments. Furthermore, assessing the interrelationship between socio-demographic factors, regional economic development, ecological dynamics, and land use practices is essential in understanding human-environmental complexities and helpful in the design of natural resource management strategies and land use policies [30,31].

This study will therefore investigate LUCC and the corresponding ecological dynamics within China's rural regions, by presenting a county experiencing depopulation in the south of China as a case study. The study will use various sources of reliable and relevant data, including remote sensing and other spatial data, to investigate and describe the land use transition processes with an emphasis on farmland within the case study region. This investigation will help to understand the transition drivers and the correlation between regional ecological dynamics and its various contributors (e.g., depopulation, afforestation, and climate fluctuations). It is expected that the results will provide new insights into local social development and ecological dynamics, which help support local decision making and will provide more realistic scenarios for planning processes. Ultimately, the findings derived from the case study are expected to provide some practical implications for sustainable land use practices in southwest China.

## 2. Study Area, Data and Methodology

### 2.1. Study Area

The study area, Rong County (29°27′ N, 104°25′ E), is located in Sichuan Province of China (Figure 1)—with an area of 1605 km². It has mixed geomorphic features including mountains, hills, and gullies, with an average elevation of 356 m. The terrain of this region displays a geomorphic transition as it gradually descends from the mountainous north-eastern area (a maximum elevation of 903 m) to the north-western and southern area (a minimum elevation of 292 m).

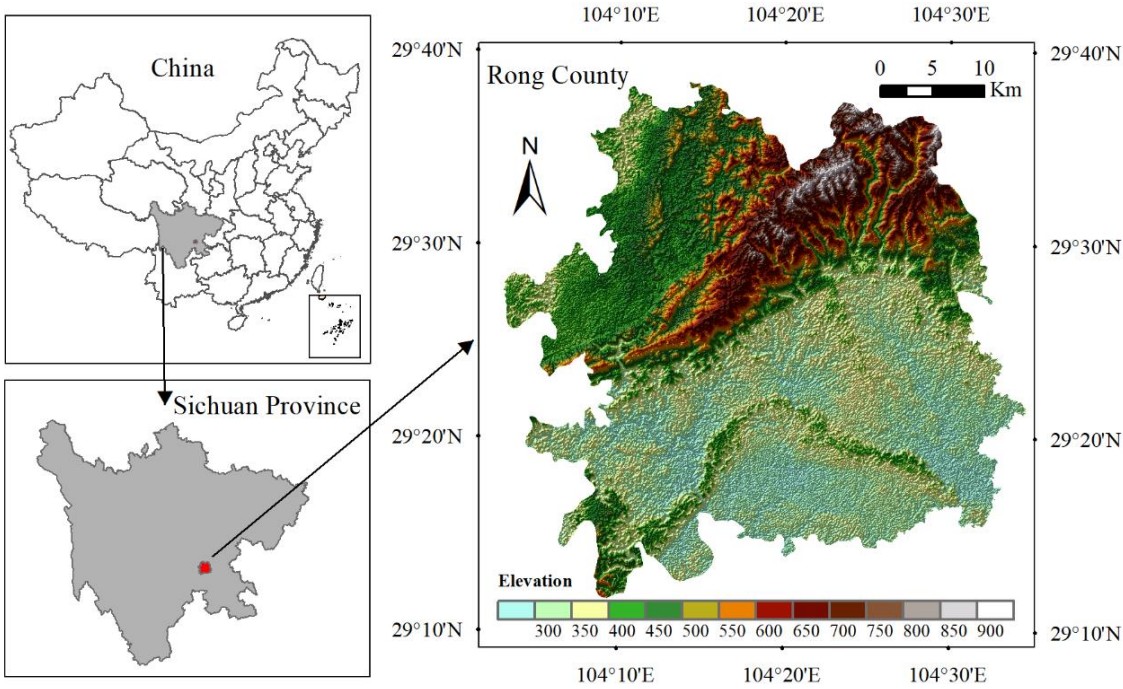

**Figure 1.** The location and elevation distribution of Rong County.

Rong County represents the typical characteristics of rural Southwest China. It has a subtropical monsoon climate, with a long frost-free period annually, abundant precipitation, and distinct seasons. Annual mean precipitation during 2000–2015 was 1012.9 mm [32]. The annual mean temperature during 2000–2015 was 17.8 °C, with the highest average temperature of 26.7 °C in August and lowest average temperature of 7.3 °C in January [32]. Due to relatively long growing seasons and the wet and warm climate in Rong County, crop rotation and continuous monocropping are two primary types of cropping system [33]. The farmland rotation schemes implemented mainly include seasonal and one-year rotations. Farmland patches are fragmented, which is a typical characteristic of smallholder farming agriculture. The main crops include rice, wheat, and corn, while sweet potatoes, oil-bearing plants, fruit trees and vegetables are also widely grown locally [33].

Rong County was an ideal area to address the research aim of this study for various reasons. Firstly, its agricultural activities are mainly small-scale household-based farming operations with a low mechanisation level which rely on local rural labour resources. Compared to the plain regions of China, agricultural activities in this region are more vulnerable to local labour population variation. Secondly, due to the rapid urbanisation process and rise in labour costs in China, substantial outmigration has occurred in the region (Figure 2a). Rong County has been experiencing both a shrinkage in the overall permanent population and the rural population. Its permanent population has decreased by 18.9%, from 666,000 in 2001 to 540,000 in 2018. Though the permanent population maintained a relatively stable trend between 2008 and 2016, the rural population has been experiencing a continuous decrease, from 80.6% in 2001 to 61.0% in 2018. The intensity of local agricultural activities has been affected in the context of depopulation. Local primary sector contribution to the Gross Domestic Product (GDP) showed a general downward trend, from 43.4% in 2000 to 24.2% in 2018, though there were fluctuations in 2004 and 2007, and a slightly upward trend in 2017–2018 (Figure 2b). Thus, there exists the possibility that the LULC structure in Rong County has experienced changes (e.g., reduction in agricultural land). Finally, the farmland topographic conditions in Rong County, which are dominated by fragmented patches, are representative of the land use in Southwest China. Land use transition phenomena within this region may reflect a general trend of LULC dynamics in other rural areas with similar outmigration trends in Southwest China.

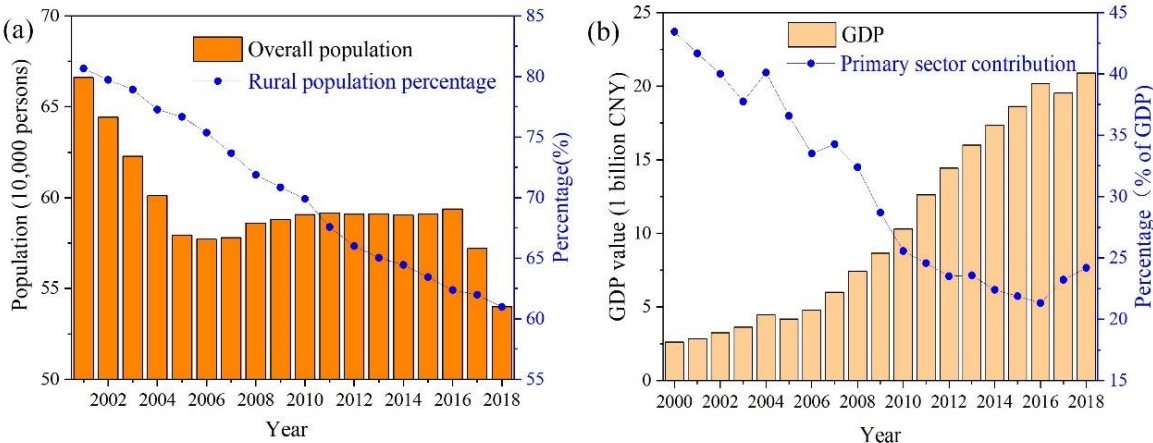

**Figure 2.** (**a**) The permanent population and rural population percentage in Rong County during 2001–2018 (source: [19]); (**b**) the time series of annual GDP (CNY 1 billion) and percentage of primary sector contribution (% of GDP) in Rong County during 2000–2018 (source: [34]).

## 2.2. Data Sources and Images Pre-Processing

This study utilised data from multiple sources, including remote sensing products, meteorological data from ground-based observatories and social-economic data from statistical yearbooks (Table 1). An integrated analysis of these data could help to gain an understanding of the changing trends in human-environment interactions within Rong County, and thus provide some implications for the rural development in southwest China.

**Table 1.** Summary of data collection for this study.

| Data Type | Time Periods | Data Sources | Usage |
|---|---|---|---|
| Landsat images | 2000, 2009, 2018 | USGS | Analysing LUCC |
| MODIS-NDVI | 2000–2018 | NASA's Earth Observing System MODIS/Terra Vegetation Indices 16-Day L3 Global 250 m | Reflecting the vegetation condition and detecting its trends |
| Climatic datasets | 2000–2015 | National Centres for Environmental Information of National Oceanic and Atmospheric Administration (NOAA) | Analysing the changes of temperature and precipitation |
| Socio-economic statistical data | 2000–2018 | Government statistical bulletins and yearbooks | Analysing the changes of socio-economic development |

This study collected about twenty 30 m-resolution Landsat images from the United States Geological Survey (USGS) Earth Resources Observation and Science Data Centre (http://www.usgs.gov). All of these Landsat images had a cloud coverage of less than 10%. This study used ENVI 5.3 to conduct radiometric calibration and FLAASH atmospheric correction for image preprocessing and used visual interpretation to observe residual clouds for further screening. Finally, 3 satellite images from Landsat 5 TM, Landsat 7 ETM+, and Landsat 8 OLI for the years 2000, 2009, and 2018, respectively, were chosen. Regional LULC is likely to have evident changes in the period of a decade. The chosen images had almost zero residual cloud cover and could clearly reveal the condition of ground features. All scenes were acquired between late April and early May, the spring season in Southeast China. This period is the crop cultivating time for farmers in the region. Choosing this period allowed the study to better distinguish farmland and other natural vegetation types (e.g., woodland and forest) as the farmland may be in the process of being sown within exposed soil, making its spectrum different from that of most natural vegetation.

The normalised difference vegetation index (NDVI) data from the Moderate Resolution Imaging Spectroradiometer (MODIS) were obtained from NASA's Earth Observing System for the period of

2000–2018. The spatial resolution of the data was 250 m. The data were collected daily and have been composited for every 16-day period following the Maximum Value Composite (MVC) rule. Monthly NDVI values were derived from two 16-day images following the MVC rule. The MVC compound algorithm can minimise the impact of cloud pollution. It can also reduce the impact of satellite sensor variations and perform reliable spectral measurements of the amount of vegetation on the surface [35]. Annual mean NDVI values were calculated by averaging monthly values filtered by a Savitzky-Golay filter [36] from January to December. The Average Value Composite (AVC) method integrates all vegetation states within a year and can more comprehensively reflect the vegetation information.

The climatic dataset, consisting of monthly mean temperature and monthly total precipitation of the study area from 2000 to 2015, were from the National Centres for Environmental Information of NOAA. Furthermore, Rong County's socio-economic data were obtained from the government statistical bulletins, the Chinese Counties Socio-economic Statistical Yearbook, and the Sichuan Statistic Yearbook.

*2.3. Research Methods*

2.3.1. LULC Classification and Transition Detection Matrix

A total of 3 high quality Landsat images were used to classify LULC in Rong County during the different historical periods. This study first collected ground observation data from different historical records in Google Earth [37] and randomly divided them into 2 groups, where about 70% of sample points were used for classification, and the remaining 30% were used as testing data for accuracy assessment of classified maps. Based on the ground observation data and relevant principles from the literature [3,38], this study considered 7 LULC types for Rong County's classification maps, including farmland (paddy), farmland (dry), forest, woodland, grassland, residential area, and water bodies. This study also incorporated various ancillary data (e.g., elevation, slope, and aspect maps) as references in the classification process to improve the accuracy of the visual interpretation.

This study employed a transition matrix to explore the conversions of LULC. This is a post classification detection technique that can detect the changing rate of each LULC types. It has been successfully used by various studies in previous LUCC research [3,39,40]. This study overlaid 3 LULC maps in ENVI 5.3 and established 3 bidirectional cross matrices. The nondiagonal entries represent the proportion of LULC that experienced a conversion from one category to another (expressed as a percentage of the total area), while diagonal entries represent the persistent proportion of that category. The row totals on the right indicate the proportion of each LULC at the initial state, and the column totals in the penultimate row indicate the proportion of each LULC at the final state. The last row represents the net gain of each LULC.

2.3.2. NDVI Trend Detection Based on Various Methods

The trend of change in the NDVI values can provide evidence for local ecological evaluation and it is complementary to the assessment of LULC transition. This study employed both parametric and nonparametric methods to perform NDVI trend analyses. In the parametric method, this study employed a linear regression model [41] to quantify change in the dependent variable, y (i.e., NDVI) against an independent variable, x (i.e., time). The magnitude of the linear model could illustrate the changing trend of NDVI during 2000–2018. This study also used an ArcMap extension (i.e., curve fit; [42]) to visualise the spatial distribution of NDVI trends. It can be used to perform a linear regression analysis on a series of raster maps of NDVI and the results present the median slope of NDVI variation for every pixel. The Mann–Kendall (MK) test [43–45] is a nonparametric test that has been extensively used for the assessment of variations of climatic trends (e.g., precipitation; [46]). It currently is more commonly applied when testing the significance of changes in NDVI trends due to its low sensitivity to missing data values, irregular data distribution, and outliers [3,47]. It can help to detect the existence of changing

trends within time series data in the absence of any seasonal variation or other cycles [46]. This method treats the values of time data series, $x_1, x_2 \ldots , x_n$ as a random sample of n independent and identically distributed variables. The trend was determined according to the test statistic Z or S, as calculated with following equations:

$$S = \sum_{k=1}^{n-1} \sum_{j=k+1}^{n} sgn\left(x_j - x_k\right) \tag{1}$$

$$sgn\left(x_j - x_k\right) = \begin{cases} +1, & if \left(x_j - x_k\right) > 0 \\ 0, & if \left(x_j - x_k\right) = 0 \\ -1, & if \left(x_j - x_k\right) < 0 \end{cases} \tag{2}$$

$$Var(S) = \frac{1}{18}\left[n(n-1)(2n+5) - \sum_{p=1}^{q} t_p\left(t_p - 1\right)\left(2t_p + 5\right)\right] \tag{3}$$

$$Z = \begin{cases} \frac{S-1}{\sqrt{Var(S)}}, & if\ S > 0 \\ 0, & if\ S = 0 \\ \frac{S+1}{\sqrt{Var(S)}}, & if\ S < 0 \end{cases} \tag{4}$$

where $x_j$ and $x_k$ are observations in year j and k (j > k), respectively, and $t_p$ is the extent of any given tier and q is the number of tiers [3].

When n is less than 10, the S test will be employed, otherwise the Z test [45]. When Z or S is greater than 0, it indicates that the time series data has an increasing trend; when Z or S is less than 0, it indicates a decreasing trend. A further hypothesis test on Z or S is needed to check whether the trend is statistically significant. If the absolute value |Z| is smaller than $Z_{\alpha/2}$ at the $\alpha$ significance level, a null hypothesis $H_0$ will be accepted, i.e., these time series data do not have a statistically significant trend; otherwise the alternative hypothesis $H_1$ will be accepted, i.e., these time series data have a statistically significant increasing or decreasing trend [3,45,48].

Sen's slope estimator is a nonparametric linear regression method [49]. It can calculate the median slope of time series data and has often been used in combination with MK test. It presents as:

$$Q_i = \frac{x_i - x_k}{j - k} \ for\ i = 1,\ 2, \ldots ,\ N \tag{5}$$

$$Q_{med} = \begin{cases} Q_{\frac{N+1}{2}}, & N\ is\ odd \\ \frac{1}{2}\left(Q_{\frac{N}{2}} + Q_{\frac{N+2}{2}}\right), & N\ is\ even \end{cases} \tag{6}$$

where $x_j$ and $x_k$ are the observed data at the year j and k (j > k), respectively; N = n (n + 1)/2, and $Q_{med}$ is the median of $Q_i$, i.e., the amplitude of annual changes, which can be transformed to relative change rate (%) by dividing the mean value of the time series ($X_{mean}$), $Q_{med}/X_{mean} \times 100$ [3].

This study used the Kendall package in the R programming to conduct the MK test for every grid within the study area and used ArcMap to map the spatial distribution of Z values. This study also calculated the trends and changing rates of the annual mean NDVI within Rong County based on the MK test and Sen's methods.

### 2.3.3. Integrated Analysis of LULC Conversions and NDVI Dynamics

LUCC could be a contributor to NDVI dynamics, such as farmland-to-woodland conversion can be beneficial to NDVI increase. However, there exists other factors (e.g., increase in vegetation canopy density, changes in crop types; [3]) that can affect NDVI performance. Conducting an integrated analysis of the LUCC and NDVI trends can be beneficial for a deeper understanding of the drivers of regional vegetation dynamics [3].

This study overlaid the postclassification detection maps and NDVI slope map within the ArcMap platform, then searched typical areas with evident NDVI change and checked if this area

also experienced LULC conversions. Based on the results of the MK test, this study also divided the areas with NDVI dynamics into 3 categories: the area with NDVI increase, the area with no evident change, and the area with NDVI decrease. This study used these categories as masks to extract the postclassification detection maps and further calculated the proportions of each LULC in each category.

This study also calculated the contribution rates of each LULC and land conversions to the total NDVI dynamics during 2000–2018. This study overlaid 2 annual NDVI maps from 2000 and 2018, respectively, within the ArcMap platform, and derived an NDVI change map by calculating their differences. Then, the polygons of various LULCs and land conversions were used as masks to extract the NDVI change map, in order to calculate their contribution rates to the NDVI dynamics. Such quantification-based analysis can help establish an integrated assessment of ecological effects with the change of quantitative structure of regional land use types. The formula are as follows:

$$
\begin{aligned}
\text{CR}_i &= \frac{\text{NDVI2018}_i - \text{NDVI2000}_i}{\text{NDVI2018}_{\text{total}} - \text{NDVI2000}_{\text{total}}} \times 100\% \text{ for i} = 1, 2, \ldots, \text{m} \\
\text{CR}_j &= \frac{\text{NDVI2018}_j - \text{NDVI2000}_j}{\text{NDVI2018}_{\text{total}} - \text{NDVI2000}_{\text{total}}} \times 100\% \text{ for j} = 1, 2, \ldots, \text{n}
\end{aligned}
\tag{7}
$$

where $\text{CR}_i$ stands for the contribution rate to the overall NDVI dynamics of the areas with land cover i and experienced no change during 2000–2018; $\text{NDVI2018}_i$ represents the total NDVI value of land cover i in 2018; $\text{NDVI2018}_{\text{total}}$ means the total NDVI value within the study area in 2018. Similarly, $\text{CR}_j$ stands for the contribution rate to the overall NDVI dynamics of the areas that experienced land conversion j during 2000–2018; $\text{NDVI2018}_j$ represents the total NDVI value of areas that experienced land conversion j during 2000–2018.

## 3. Results

### 3.1. LULC Maps and Transition Matrices

The LULC patterns of Rong County in 2000, 2009, and 2018 are shown in Figure 3. The dominant land cover types identified from the Landsat images were farmland (paddy), farmland (dry), forest, and woodland, whereas the area of grassland, residential area and water bodies were relatively small. The LULC pattern of Rong County is closely related to local topography distribution. Forest and woodland are mainly distributed on the mountains that traverse the north, as well as on the hills across the southern areas. Farmland is mainly distributed in the northwest, central, and southern periphery areas, where the terrain is relatively flat, and elevation is relatively low. Residential area within the study region has significantly increased during 2000–2018, while high-elevation farmland showed a decreasing trend and has gradually converted to vegetation.

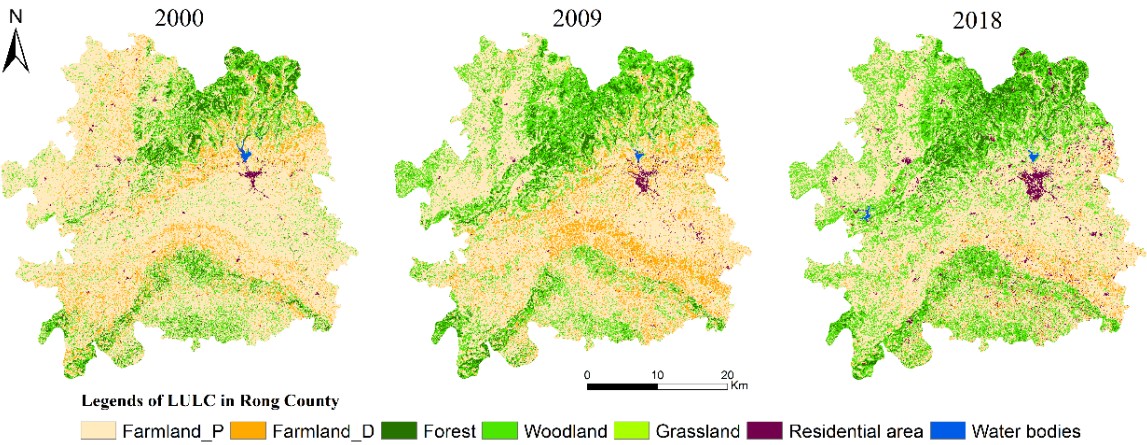

**Figure 3.** The land use and land cover (LULC) maps of Rong County in 2000, 2009, and 2018.

The transition matrix during 2000–2018 is shown in Table 2. LULC transitions from 2000 to 2018 were mainly characterised by conversions between farmland (both paddy and dry), woodland, and residential area. During 2000–2018, the most distinct conversions were conversion of farmland (paddy) to woodland (15.33% of the total area), conversion of farmland (dry) to farmland (paddy) (6.04%), and conversion of woodland to farmland (paddy) (5.14%). The LULC with the largest net increase was woodland (10.36%), while the LULC with the largest net decrease was farmland (paddy) (−9.21%). The net growth of residential area accounted for only 2.09% of the total area, but it expanded by 268.34% compared to 2000. It is the fastest growing LULC. The area of grassland and water bodies remained almost the same.

**Table 2.** Percentages of conversion of each LULC during 2000–2018 (% of total area).

| Year | | 2018 | | | | | | | Total (2000) |
| --- | --- | --- | --- | --- | --- | --- | --- | --- | --- |
| | | FaP | FaD | Fot | Wod | Gras | Res | WB | |
| | FaP | 45.31 | 3.27 | 0.63 | 15.33 | 0.17 | 1.67 | 0.04 | 66.42 |
| | FaD | 6.04 | 0.62 | 0.08 | 1.49 | 0.08 | 0.41 | 0.00 | 8.72 |
| | Fot | 0.35 | 0.00 | 2.60 | 1.36 | 0.00 | 0.09 | 0.00 | 4.40 |
| 2000 | Wod | 5.14 | 0.28 | 2.30 | 10.60 | 0.02 | 0.26 | 0.02 | 18.62 |
| | Gras | 0.00 | 0.02 | 0.00 | 0.17 | 0.69 | 0.04 | 0.00 | 0.92 |
| | Res | 0.32 | 0.01 | 0.01 | 0.03 | 0.00 | 0.40 | 0.00 | 0.78 |
| | WB | 0.05 | 0.00 | 0.00 | 0.00 | 0.00 | 0.00 | 0.08 | 0.14 |
| Total (2018) | | 57.22 | 4.20 | 5.62 | 28.98 | 0.96 | 2.87 | 0.14 | 100.00 |
| Net gain in 2018 | | −9.21 | −4.52 | 1.22 | 10.36 | 0.04 | 2.09 | 0.00 | |

Notes: FaP, farmland (paddy); FaD, farmland (dry); Fot, forest; Wod, woodland; Gras, grassland; Res, residential area; WB, water bodies.

The transition matrices during 2000–2009 and 2009–2018 are shown in Tables 3 and 4, respectively. LULC transitions were mainly characterised by conversions between farmland (both paddy and dry) and woodland. In both periods, the most distinct conversion was the conversion of farmland (paddy) to woodland (9.61% in 2000–2009, and 11.79% in 2009–2018, respectively). The LULC with the largest net increase was woodland during 2009–2018 (7.85%). The LULC with the largest net decrease was farmland (dry) during 2009–2018 (−6.53%). Most LULC types maintained the same trend over both periods. For example, farmland (paddy) decreased over both periods; while forest, woodland and residential area increased in both periods. Only farmland (dry) and water bodies had conversed trends in two periods.

**Table 3.** Percentages of conversion of each LULC during 2000–2009 (% of total area).

| Year | | 2009 | | | | | | | Total (2000) |
| --- | --- | --- | --- | --- | --- | --- | --- | --- | --- |
| | | FaP | FaD | Fot | Wod | Gras | Res | WB | |
| | FaP | 47.96 | 7.88 | 0.42 | 9.61 | 0.00 | 0.54 | 0.00 | 66.42 |
| | FaD | 5.42 | 2.20 | 0.03 | 0.72 | 0.25 | 0.10 | 0.00 | 8.72 |
| | Fot | 0.56 | 0.00 | 2.68 | 1.15 | 0.00 | 0.01 | 0.00 | 4.40 |
| 2000 | Wod | 6.43 | 0.46 | 2.16 | 9.52 | 0.03 | 0.04 | 0.00 | 18.62 |
| | Gras | 0.00 | 0.15 | 0.00 | 0.09 | 0.66 | 0.01 | 0.00 | 0.92 |
| | Res | 0.43 | 0.04 | 0.01 | 0.02 | 0.01 | 0.26 | 0.00 | 0.78 |
| | WB | 0.05 | 0.01 | 0.01 | 0.00 | 0.00 | 0.01 | 0.07 | 0.14 |
| Total (2009) | | 60.85 | 10.73 | 5.30 | 21.13 | 0.96 | 0.96 | 0.07 | 100.00 |
| Net gain in 2009 | | −5.57 | 2.01 | 0.90 | 2.51 | 0.04 | 0.18 | −0.07 | |

Notes: FaP, farmland (paddy); FaD, farmland (dry); Fot, forest; Wod, woodland; Gras, grassland; Res, residential area; WB, water bodies.

**Table 4.** Percentages of conversion of each LULC during 2009–2018 (% of total area).

| Year | | 2018 | | | | | | | Total (2009) |
|------|------|------|------|------|------|------|------|------|------|
| | | FaP | FaD | Fot | Wod | Gras | Res | WB | |
| | FaP | 45.10 | 1.95 | 0.52 | 11.79 | 0.09 | 1.35 | 0.05 | 60.85 |
| | FaD | 7.06 | 2.10 | 0.01 | 0.72 | 0.11 | 0.73 | 0.00 | 10.73 |
| | Fot | 0.28 | 0.01 | 3.06 | 1.88 | 0.00 | 0.07 | 0.00 | 5.30 |
| 2009 | Wod | 4.32 | 0.06 | 2.03 | 14.45 | 0.02 | 0.22 | 0.03 | 21.13 |
| | Gras | 0.00 | 0.03 | 0.00 | 0.12 | 0.73 | 0.07 | 0.00 | 0.96 |
| | Res | 0.45 | 0.06 | 0.00 | 0.02 | 0.00 | 0.42 | 0.00 | 0.96 |
| | WB | 0.01 | 0.00 | 0.00 | 0.00 | 0.00 | 0.00 | 0.06 | 0.07 |
| Total (2018) | | 57.22 | 4.20 | 5.62 | 28.98 | 0.96 | 2.87 | 0.14 | 100.00 |
| Net gain in 2018 | | −3.62 | −6.53 | 0.32 | 7.85 | 0.00 | 1.91 | 0.07 | |

Notes: FaP, farmland (paddy); FaD, farmland (dry); Fot, forest; Wod, woodland; Gras, grassland; Res, residential area; WB, water bodies.

In general, the land transition process in the study area was characterised by the conversion of farmland (both paddy and dry) to residential area and woodland. The expansion of residential area reduced low-elevation farmland, while land conversions at relatively high elevation were mainly from farmland to woodland and forest. It was noteworthy that farmland (dry) and water bodies showed conversed trends over the two periods.

### 3.2. Vegetation Changing Trend Analysis Based on NDVI

The changing trends of NDVI in Rong County from 2000–2018 showed distinct spatial disparities. Figure 4a shows the result of the MK test of annual NDVI performance. Grids with decreased and nonchanging NDVI trends were mainly concentrated in the east and southeast regions of Rong County. The areas with the most significant decreased NDVI values were always in the central eastern region, where the local township was located. NDVI values of a few broken patterns located in the northeast periphery also experienced decreasing trends. Grids with increased NDVI trends cover most areas except the east and southeast parts in Rong County. Overall, grids with statistically significant ($Z \geq 1.64$) positive NDVI trend accounted for 83.71% of the land surface, and grids with no significant change ($-1.64 < Z < 1.64$) covered 15.73% of the land surface, while statistically significant ($Z \leq -1.64$) negative NDVI trends covered only 0.56% of the land surface.

Figure 4b shows the results of MK test Z value, Sen's slope, and linear regression analysis of the NDVI changing trend in Rong County. The Z value was 3.29, representing the increase of annual NDVI value exceeded the confidence level of 99.9% (Z = 3.27). This means that local annual NDVI increase was a statically significant increase. The Sen's slope estimation of annual NDVI was 0.0062 per year. This represented the median slope of annual NDVI increase during 2000–2018 and was slightly different from the slope calculated by the regression analysis (0.0059 per year). During 2000–2009, the observations fitted well with the trendline of the regression analysis, while there were relatively more fluctuations in the observations during 2010–2018.

### 3.3. Contribution of LUCC to NDVI Dynamics

Figure 5 shows the spatial distribution of annual NDVI changing slope in Rong County. This study defined the annual NDVI changing slope into six categories based on local ecological dynamics: significant (above 0.01), strong (0.0075–0.01), moderate (0.005–0.0075), and slight increases (0–0.005); and moderate (−0.01–0) and significant decreases (below −0.01). Areas with significant, strong, moderate, and slight NDVI increases accounted for 4.56%, 24.51%, 38.48%, and 30.17% of the total area, respectively. Areas with moderate and significant NDVI decreases only accounted for 2.10% and 0.20% of the total area, respectively.

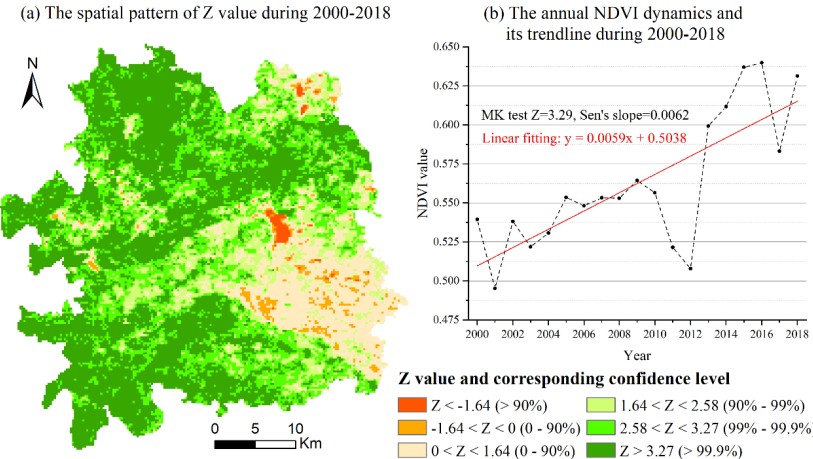

**Figure 4.** (**a**) The spatial pattern of the Z values and related confidence levels. When Z > 0, an increase trend could exist; while when Z < 0, a decrease trend could exist. Normally, when a grid has a confidence level higher than 90%, it can be seen as statistically significant, meaning the grids changing trend was credible. However, MK test cannot show the extent of change of each grid, and a high confidence level does not mean drastic change. (**b**) The changing trends of annual normalised difference vegetation index (NDVI) values of Rong County during 2000–2018.

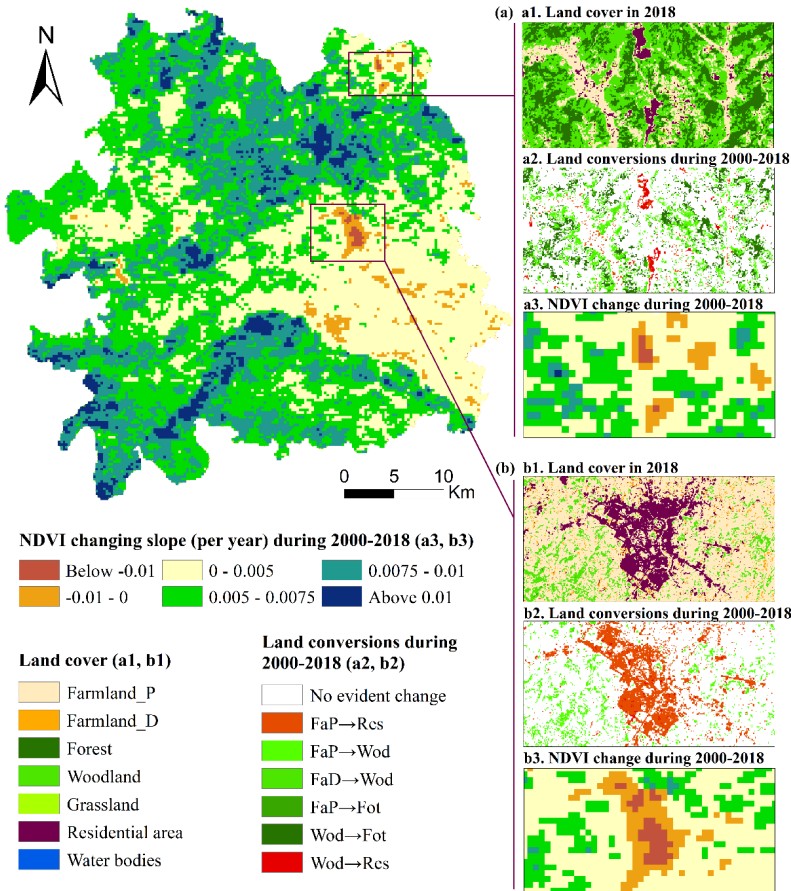

**Figure 5.** The spatial distribution of annual NDVI changing slope during 2000–2018; (**a**) the comparison between the patterns of land cover, land conversions and NDVI change for a rural residential region; (**b**) the comparison between the patterns of land cover, land conversions and NDVI change for the urban area (Notes: The same as Table 2).

This study compared the elevation distribution map, LULC maps and the NDVI changing rate map and found that: (1) areas with significant, strong, and moderate NDVI increases were mainly distributed in regions with relatively higher elevation and; (2) areas with NDVI decreases were mainly concentrated in residential areas and reservoirs, and some farmlands in southeast regions that are close to local road networks also showed decreased NDVI trends. Especially, the patterns of areas with significant NDVI decreases and surrounding regions were consistent with the areas that experienced conversions from other LULC types to residential areas (Figure 5a,b). However, this study didn't find evident spatial correlations between the patterns of areas with increased NDVI and land conversions. A quantitative analysis was therefore conducted to further explore the contribution of LULC and land conversions to NDVI dynamics.

The composition of areas with different NDVI trends during 2000–2018 was calculated (Figure 6a). Firstly, farmland and the conversion from farmland to residential area both accounted for 34.03% of the area with decreased NDVI. Both had spatial correlations with human activities (Figure 5). Secondly, areas with no evident NDVI dynamics were mainly concentrated on farmlands (59.32%). The mutual conversion between some LULCs with similar spectra (e.g., paddy land and dry land) also did not bring about evident changes in NDVI. Finally, land with no conversions accounted for about 58.25% of the areas with increased NDVI, where farmland (42.29%) and woodland (12.05%) were the two main compositions. Land conversions (e.g., farmland-to-woodland conversion, 17.49%) occupied relatively fewer areas.

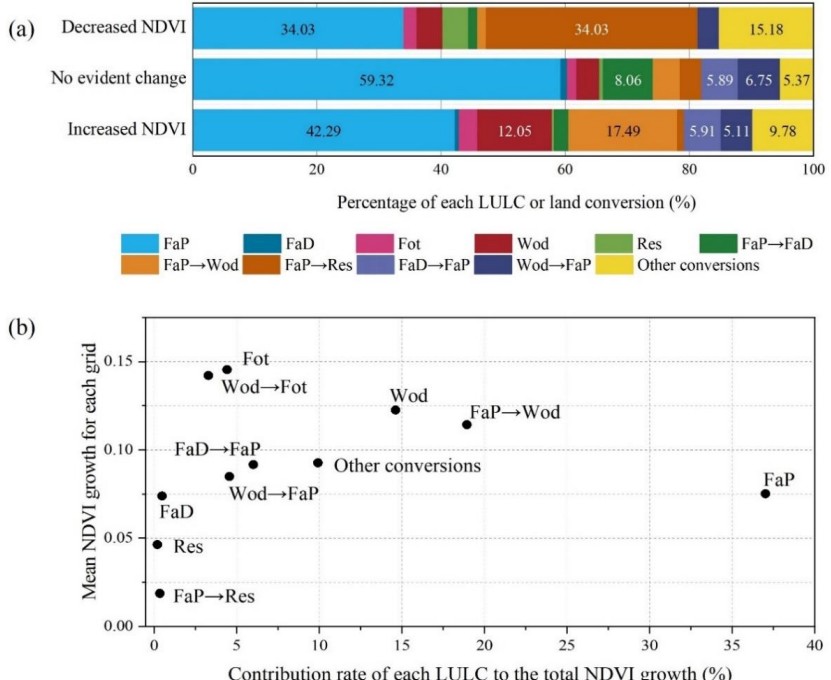

**Figure 6.** (**a**) The percentage of each LULC and some typical land conversions in areas with different NDVI trends during 2000–2018; (**b**) the mean NDVI growth for each grid, and the contribution rate to the total NDVI growth of each LULC and some typical land conversions during 2000–2018 (notes: The same as Table 2).

The contribution rates of LULC and land conversions to NDVI dynamics during 2000–2018 were calculated (Figure 6b). Farmland made the highest contribution (37.02%) to the local NDVI growth due to its wide distribution, but its NDVI growth per grid (0.075) was medium. Farmland-to-woodland conversion and woodland also made contribution of 18.94% and 14.62%, respectively. Forest and woodland-to-forest conversion had the highest mean NDVI value per grid (0.145 and 0.142, respectively), but their contributions were relatively low as their areas were small. Residential area and the conversion from farmland (paddy)

to residential area had both the lowest mean NDVI growth (0.046 and 0.018, respectively) and the rarest contribution.

## 4. Discussion

### 4.1. Dynamics in Each LULC Type and Driving Mechanisms

The expansion of residential area in Rong County occurred in both urban and rural regions, which was inevitable given the rapid urbanisation currently being experienced in China. Local urbanisation rate has directly led to a sprawl in local urban settlements. Urbanisation and industrialisation also improved the living standard of local residents, which may contribute to the expansion of rural settlements in spite of the depopulation. The decreased farmland (both paddy and dry) could be the result of a combination of urbanisation, rural depopulation, and the attendant reduction in the rural labour force. On the one hand, the expansion of urban areas would occupy the land around the city, directly leading to the conversion of farmland to residential area. From 2000 to 2016, the per capita area of farmland decreased from 0.094 ha to 0.086 ha in China [50] due to urbanisation and industrial development. On the other hand, there was a contraction in both the overall population and rural population proportions in Rong County (Figure 2). Figure 7a shows an overall decreasing trend in the local population density over the period 2000–2015, with a more rapid decrease in rural population density, and a relatively stable but still decreasing trend in urban population density. This shows that the local population tended to migrate outward, and those who stayed behind chose to gather more in the local urban areas. The average annual emigration rate within Rong County was 7.39‰ during 2010–2018, which was always higher than the migration rate (2.54‰ per year; Figure 7b). In particular, in 2017 and 2018, the net emigration rate had a substantial increase, and the net emigration rate peaked (8.66‰) in 2018 (Figure 7b). Figure 7c shows that the proportion of people older than 60 years had a continuous increase from 17.30% in 2010 to 23.95% in 2018, while people in other age groups all showed decreases between 2010 and 2018. These could have an impact on local rural livelihoods and the availability of local labour force. The China County Statistical Yearbook pointed out that the number of local rural labour force engaged in local agriculture, forestry, animal husbandry, and fishery reduced by 35.2%, from 281,305 in 2000 to 181,682 in 2012 [51]. In this context, local household-based agricultural activities could have been affected.

The characteristics of agricultural activities in Rong County are similar to most regions in Southwest China, including a high dependence on rural labour force, small-scale farming operations, and a low mechanisation level, which could limit local agricultural economic viability compared to the east plain regions of China [28]. The farming patterns in this region were vulnerable to local labour population variations and could be easily affected. This declined intensity of local agricultural activities could further increase the population outflow due to a decrease in local agricultural employment opportunities. A decrease in local farmland area was then inevitable, given the outflow of rural population diminished the availability of labour to serve local low-mechanisation agriculture, thus parts of the farmland may have been abandoned, resulting in the conversion of farmland to other land cover.

The increased woodland could be attributed to farmland abandonment, adjustments in local agricultural activities, and afforestation. Farmland abandonment refers to the cessation of management and interventions on farmlands [53], resulting in the degradation of farmland facilities to the extent that they cannot be easily used again [54], and can be accompanied by a degree of vegetation restoration [55]. Wang et al. pointed out that farmland abandonment is a prevalent phenomenon in the southwestern regions of China, where rewards from the agricultural industry are relatively low compared to the eastern regions [56]. The secondary succession of vegetation is one of the most evident consequence of farmland abandonment [57]. Studies from Europe have indicated that vegetation succession on abandoned farmland can be rapid, and can occur within less than 5 decades, where an abandoned field can become a dense forest [58,59]. A study from Nepal indicated that a large proportion of rain-fed farmlands on hilly terrains can transition into woody shrubland within a decade [60]. The research period for this study covered 19 years, which is a relatively short evolutionary cycle more likely

leading to an increase in woodland rather than dense forest. Apart from farmland abandonment, another factor contributing to the farmland-to-woodland conversion could be an adjustment in the local agricultural structure. The yield of cash crops increased over the years, while the yield of food crops remained at a relatively stable level (Figure 8a). Expansion of cash crop cultivation could contribute to local ecological performance, as the spectrum of vegetables and fruit trees were different from those of crops. The cash crops could have been classified as grassland or woodland rather than farmland, even though they may actually still be used for economic agricultural activities. Finally, Rong County has sustained an active afforestation. During 2008–2018, the average annual afforestation area was 23.88 km$^2$ and the cumulative afforestation area was 262.68 km$^2$ (Figure 8b), mainly aiming at planting economic and fast timber forests. However, the impacts of these afforestation activities on local farmland-to-woodland conversion should not be overestimated, given most afforestation occurred within the high-elevation areas, where the amount of farmland was relatively low. Furthermore, the local annual timber harvest was also large (22,650 m$^3$ per year; [33]), thus the net gain in woodland area brought by local afforestation may not be high.

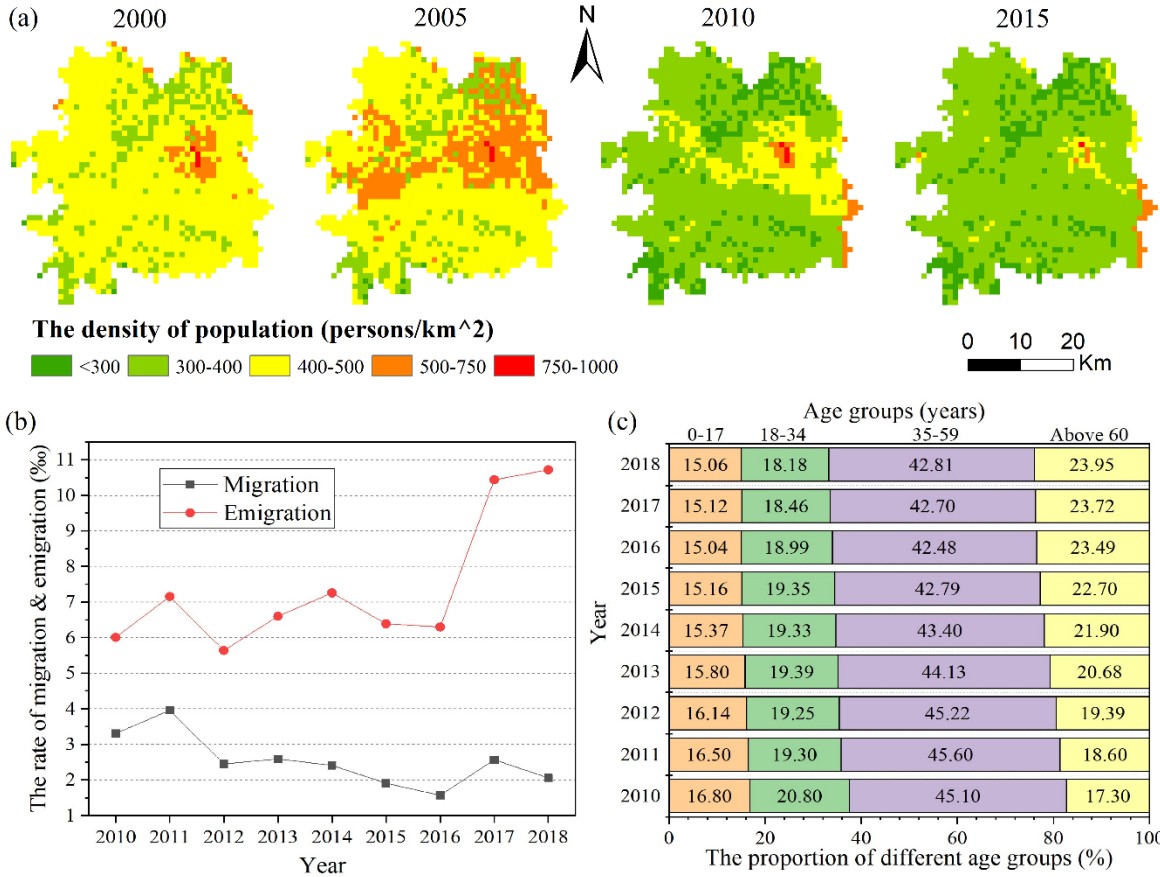

**Figure 7.** (**a**) The spatial distribution map of the population density of Rong County in 2000, 2005, 2010, and 2015 (source: [52]); (**b**) the annual rate of migration and emigration in Rong County during 2010–2018 (source: [33]); (**c**) the annual age structure of the population within Rong County during 2010–2018 (source: [33]).

## 4.2. NDVI Changing Trends and Drivers

NDVI in the study area showed an upward trend. The average annual NDVI values and all average seasonal NDVI values in the study area were higher during 2010–2018 than that during 2000–2009. The MK test revealed that areas with increased NDVI values comprised of 97.5% of the total area, while 83.7% of the total area was statistically significant ($p \leq 0.1$) positive. This indicates that

the vegetation condition in the study area improved during 2000–2018. This finding is consistent with the results of several other studies that also found positive NDVI trends in Southern China [61–63].

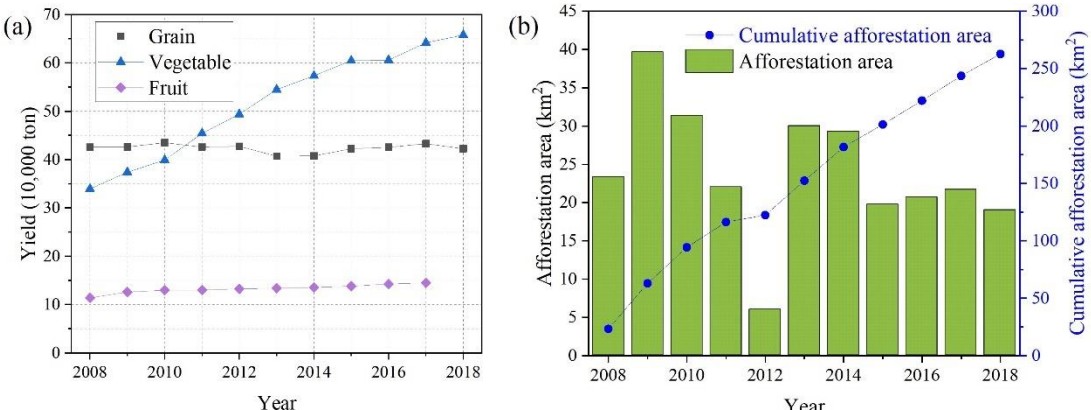

**Figure 8.** (**a**) The annual yield of grain, vegetable and fruit during 2008–2018; (**b**) the annual afforestation area and the cumulative afforestation area during 2008–2018 (source: [33]).

Local land use conversions had both positive and negative impacts on local NDVI dynamics. Figure 6 shows that decreased local NDVI was mainly due to the expansion of the residential area and the spectra change of some farmlands. This part of farmland was close to local road networks and rural settlements. This indicates that human activities can directly affect their surrounding areas and change vegetation conditions. Judging from the overall situation in relation to local vegetation, the negative impact of human activities was not large. Only 2.5% of area showed a statistically significant decrease in NDVI. Furthermore, Figure 6 also shows that land conversions (e.g., farmland-to-woodland conversion) could have positive effects on local NDVI increases, but this was not the main contributor for such a distinct vegetation improvement within the study region.

This study found that some regions did not experience evident land use conversions but did have a strengthened NDVI. This indicates that the improvement of the local vegetation condition may be caused by the combination of land conversion and the enhancement of the canopy density of the existing vegetation. To comprehensively investigate the drivers behind this increased NDVI trend, this study adopted the Pearson correlation analysis from the literature [3,64]. This study analysed the Pearson correlation between the annual NDVI and the local annual permanent population, the annual afforestation area, the annual overall precipitation, and the annual mean temperature.

Depopulation could have correlation with regional vegetation conditions [65,66]. Van der Geest et al. found a significant but weak correlation between migration and vegetation cover in Ghana [64]. The Pearson correlation between the permanent population dynamics in Rong County and annual NDVI trend during 2000–2018 was −0.514 and passed the significance test ($p < 0.01$). This indicates a moderate and statistically significant negative correlation between the depopulation in the study area and the increased NDVI trend. Some studies have analysed how rural outmigration processes affected the regional ecological environment in Southwest China [28,61]. Among these outmigration processes, the outflow of the working aged population (Figure 7c) played a role in regional vegetation restoration [61,67]. Negative disturbances in the ecological environment (e.g., firewood gathering) may be reduced as a result of the outmigration of male labour forces, as firewood gathering is a time-consuming and labour-intensive activity that is more commonly undertaken by working aged residents in rural regions [61,68]. Furthermore, rural outmigration could change the long-standing management practices of rural households in relation to natural resources, and vegetation improvement could be a possible result.

The outflow of peasant workers will not only directly weaken the labour available for agriculture, but also have an indirect impact, which is attributed to China's unique household registration system. The peasant migrants can still have the ownership of their rural houses and farmlands after moving to the cities, or have family members left in the origin areas [12]. The out-migrated workers have "left the

soil but not separated from their villages" [12], which can bring home higher income to their families. An investigation of rural households in Southwest China indicated that the income of out-migrant households was higher than that of households with no or less out-migrants [69]. This is because out-migrated workers in China usually send part of their income to their rural families, which is usually higher than the returns from farming. The increased income may reduce the dependency of out-migrant households on land [64] and result in the abandonment of sloping farmland [61]. Olsson et al. believed that widespread vegetation increase, due to reduced areas under cultivation, could be a possible result of increased rural-to-urban migration and increased dependence on remittances [66]. Yang et al. held a similar opinion that the dependency of migrant households in Southwest China on conventional agricultural activities (e.g., cropping and livestock farming) was much lower [70]. Furthermore, rural households in Southwest China were usually dependent on biomass fuels to meet energy requirements, and firewood is the most commonly used and accessible biomass fuel [71]. Higher incomes within migrant households may promote a switch from firewood to cleaner and more efficient fuels. This may reduce the negative impact of human disturbances on forests and contribute to forest restoration in the study area. Other factors related to outmigration, such as livelihood diversification and more sustainable farming techniques, may also contribute to forest restoration [61].

Continuous afforestation activities in the study area (Figure 8b) had a positive but weak significance for NDVI growth. The Pearson correlation between annual mean NDVI and annual afforestation area during 2008–2018 was 0.149 ($p < 0.01$), suggesting a weak but statistically significant correlation. Afforestation is conducive to forest restoration and involves more than simply increasing the forest area. The afforestation of marginal land can change the configuration of the forest landscape and increase the soil water capacity, which has benefits for vegetation growth [61]. However, it is worth mentioning that the effects of afforestation on local ecological restoration was limited, as afforestation activities in Rong County are mainly concentrated within economic and timber forests, where logging activities occur each year [33]. Its vegetation improvement was more attributed to the vegetation foundation and natural conditions, rather than external influences. Thus, the cumulative afforestation area and its contribution to NDVI should not be overestimated.

Regional climate had positive correlations with NDVI performance. The Pearson correlation between annual NDVI trend and annual overall precipitation and average temperature during 2000–2018 was 0.089 ($p < 0.01$) and 0.505 ($p < 0.01$), respectively. Numerous studies have claimed that climate change can have impacts on NDVI dynamics [3,28,61,72]. Vegetation growth is closely connected to climate variables (e.g., temperature and precipitation) and serves as an indicator of climate change [72]. Han and Song pointed out that interannual variations of phenological features could not be entirely attributed to changes in land-use types but might also be the response to climatic disturbances [28]. However, both the increasing trends of precipitation and temperature within the study region were not significant (Figure 9). The $R^2$ values (0.04 for the precipitation and 0.03 for the temperature) are statistically negligible. The climatic impacts within the study region were long-term and cumulative, and the Pearson correlation showed that the mild temperature had a relatively more significant contribution.

To conclude, the trend of land conversion and NDVI change indicated that the ecological environment of Rong County was generally on an optimistic and positive trend. Local ecological improvement is self-regulating in behaviour with the ecosystem under the combined influence of rural outmigration and a climate warming trend. This study also found human activities were the main driver of local vegetation degradation. However, the areas with decreased NDVI only occupied about 2.5% of the total area, indicating that human activities did not have a significant negative impact on the ecological environment.

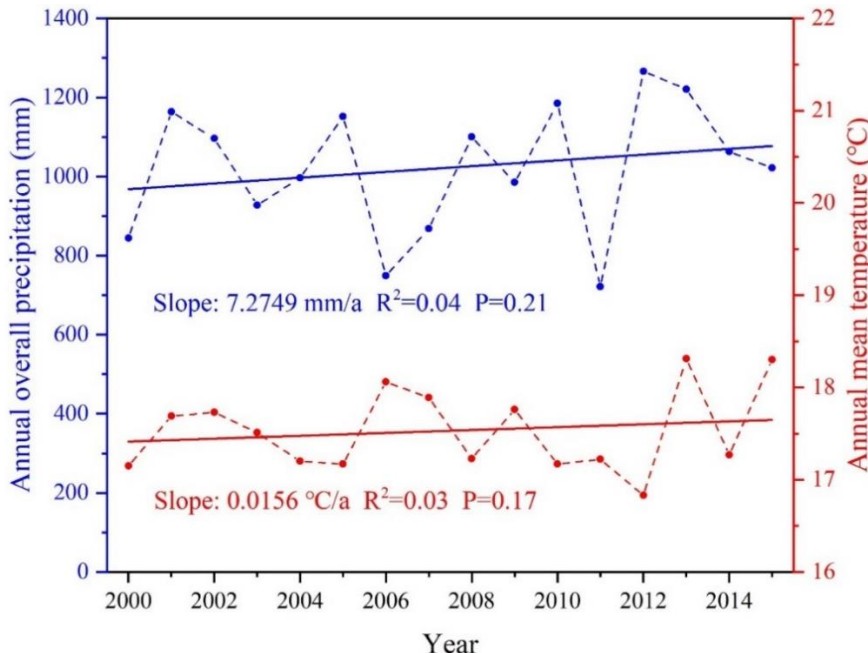

**Figure 9.** Annual average temperature and overall precipitation and their trendlines in Rong County during 2000–2015 (source: [32]).

*4.3. Implications for the Rural Development in Southwest China*

The social and ecological dynamics of Rong County has implications for rural development in Southwest China. Rong County represents the typical characteristics of rural Southwest China, with a mild to warm and wet climate, moderate to high natural vegetation cover, and decreasing trends in arable land and labour force. The problems and dynamics it encountered in the development process might also be faced by other rural areas in Southwest China, suggesting there are several implications that can be drawn from the case study.

Abandonment on high-slope farmland can contribute to ecological restoration and has a relatively small impact on regional agriculture. Farmland abandonment in Rong County mainly occurred in the hilly areas, which were more suitable for natural conservation and regeneration of vegetation, rather than crop cultivation. The ecological repercussions of abandoned farmland are affected by various factors, including the local biotic and abiotic environment, historical agricultural activities, type, duration, and intensity [73]. There exists a consensus that old field succession can affect regional habitats, ecosystem services, soil fertility, and biodiversity [73]. Some studies claimed that land abandonment could be a chance for forest regeneration and enhancement of carbon sequestration, and can benefit woodland birds [2,28,73–75]. In the context of urbanisation and the rural exodus in Southwest China, these biophysically poor arable lands have often been spontaneously abandoned [28]. Han and Song suggested that farmers should be encouraged to reduce unsustainable agricultural activities in mountainous areas with steep slopes [28]. As local agricultural yields weren't significantly threatened, it might be a good choice to allow those high-slope farmlands to undergo forest succession. Simultaneously, correlation analyses showed that the increase in local vegetation was more due to lack of human management and climate warming than ecological construction. It then may not be necessary to extend ecological construction projects (e.g., Green to Grain Project) in the region as it would need to provide additional subsidies to farmers.

Farmers in the southwest of China should make full use of the existing low-slope arable land and curb the declining trend to meet the challenges brought on by urbanisation. Typically, urbanisation is thought to put additional pressure on regional arable land and water resources [76]. For every 1% increase in urban population, an additional 140 thousand ha of arable land in China needs to be claimed to feed the population [77]. On the other hand, urban sprawl tends to take up

a large portion of high-quality arable land that have medium slope and short spatial distances to settlements. How to deal with the dual pressures brought about by urbanisation (i.e., increased food demand and deceased high-quality arable land) is an inevitable challenge in the urbanisation progress in Southwest China. To mitigate the impacts and make the most of existing farmland, agricultural infrastructure should be enhanced, such as constructing more water conservation facilities and road infrastructure. Centralisation and mechanisation of agriculture should also be strengthened, which may improve the productivity of agricultural activities in spite of the shortages in rural labour. Furthermore, improved crop varieties and utilisation of fertiliser can help maintain the local grain yield. The International Rice Research Institute stated that the average yield of rice in China had improved to 6.5 tons per hectare, the highest in Asia [78]. Rising unit yields of grains can allow farmers to use less labour to maintain total production at the same level as before. Local governments can also introduce policies and public education activities to guide farmers towards scientific agricultural activities, in order to efficiently use high-quality farmland instead of reclaiming high-slope farmland. This may help to sustainably manage local land use and be conducive to the protection or regeneration of the regional ecological environment.

### 4.4. Future Research Perspectives

This study contributed to a better understanding of how the regional land use transition and the corresponding ecological dynamics may occur under the impact of urbanisation and population outflow based on a case study in Southwest China. The study has demonstrated the intrinsic links between land conversion and vegetation dynamics, producing good model fits. However, there still exist areas where improvements can be made in the future.

This was a micro-scale study and the study area was only about 1605 km$^2$. Although the study area has the typical characteristics of Southwest China, the conclusions drawn from this study could have contingency and uncertainty due to the micro scale. In future, studies with larger scales and longer time series could be considered. It would reflect a more general and persuasive trend.

Research on social factors should be reinforced in subsequent studies. Existing models can be optimised by considering household-level decisions and field investigation. This study considered the links between ecological, demographic and agricultural factors, and land conversion processes. However, the spatial and temporal changes in land use can also be influenced by household-level decisions, such as institutional conditions (e.g., land use rights, agricultural subsidies), agricultural markets (e.g., agricultural commodity prices), and farming characteristics (e.g., part-time farmers, nonagricultural income) [28]. These factors are usually collected via household surveys during the study. When conducting similar research in the future, consideration should be given to conducting a household survey of farmers and combining the survey data with official data to provide greater veracity to the findings. In addition, field visits and family surveys can be conducive to building an accurate understanding of the study area. Detailed ground truth data can be collected in field visits and used for comparison and verification with high-accuracy data provided by Google Earth.

This study demonstrated that NDVI trend analysis and LUCC analysis may be complementary. Such an integrated analysis may help to establish the mechanisms behind the processes of LULC transition and vegetation dynamics [79]. It is considered that further research would benefit by concentrating on the establishment of mathematical coupling models between LUCC and other vegetation indices dynamics, with an emphasis on the quantification of the diverse effects of LUCC on vegetation indices change. A number of algorithms or products have been developed to obtain surface parameters over long time spans, including land surface temperature, soil moisture, and evapotranspiration volume [3]. Therefore, in future studies other surface parameters may be included to monitor vegetation variation. Such remote sensing research needs to be combined with field investigation, regional socioeconomic statistics, case studies, and mechanism studies. In order to enhance the potential of remote sensing research in environmental monitoring and natural resource management, further effort should also be made to ensure field investigations and comparative

case studies at multiple scales. This could help deepen the understanding of the intrinsic connections between human activities, LUCC, and vegetation dynamics.

## 5. Conclusions

This study assessed the outmigration-related land use transition process of a county in Southwest China during 2000–2018 and corresponding ecological dynamics based on various sources of data, including remote sensed, climatic, socio-economic, and agricultural data. This study investigated how the local land use patterns, and agricultural, social, and ecological environments were changing in the context of depopulation and found that:

Local land conversion process was mainly characterised by the conversion of farmland (−18.3%) to residential area (+268.3%) and woodland (+55.6%) during 2000–2018. Urbanisation directly led to local residential area expansion. The decreased farmland (both paddy and dry) could be the result of a combination of urban sprawl, rural depopulation, and the attendant reduction in the rural labour force. Combined with local small-scale household-based farming, part of farmland may have been abandoned and vegetation succession occurred. Other reasons, including the increase in cultivation area of vegetable and fruit trees, and economic afforestation, could also contribute to the reduction of local farmland and vegetation growth.

Local ecological environment was on a positive trend. About 83.7% of the area had a statistically significant increase of NDVI. Annual NDVI has been increasing at a rate of 0.0062 per year. Areas with negative NDVI only occupied 2.5% of total area and was directly caused by human activities. The ecological improvement was of a self-regulating behaviour of the ecosystem under the combined influence of rural outmigration and a climate warming trend. This study found moderate and statistically significant correlations between increased NDVI trend and depopulation (R = −0.514, $p < 0.01$) and local mild temperature (R = 0.505, $p < 0.01$). Decreased human activities in rural areas and the remittances provided by migrant workers were conducive for vegetation regeneration. The effect of climate change on local NDVI improvement was long-term and cumulative, but vegetation responded quickly when there were abrupt fluctuations in climate. The effect of local afforestation on NDVI change was weak.

The case study has implications for the rural development of Southwest China. It could be a good choice to allow those high-slope farmlands to undergo forest succession rather than cultivation. Farmers in Southwest China should make full use of the existing low-slope arable land and curb the declining trend to meet the challenges brought on by urbanisation. Enhanced agricultural infrastructure, mechanised farming and guide from local government can help achieve this goal. Future research perspectives include conducting a larger-scale research, conducting field investigation, considering household-level decisions, and conducting an integrated analysis of LUCC and other vegetation indices.

**Author Contributions:** Conceptualisation, T.X. and Y.Y.; methodology, Y.Y.; software, Y.Y. and T.W.; validation, Y.Y.; formal analysis, Y.Y.; investigation, Y.Y.; resources, Y.Y. and T.X.; data curation, Y.Y. and T.W.; writing—original draft preparation, Y.Y.; writing—review and editing, Y.Y. and T.W.; visualization, Y.Y. and T.W.; supervision, T.X.; project administration, T.X. All authors have read and agreed to the published version of the manuscript.

**Funding:** This research was supported by the Fenner School of Environment and Society's Student Research Project Allocation, at the Australian National University.

**Acknowledgments:** We express our gratitude to Dona Whiley for her professional editing.

**Conflicts of Interest:** The authors declare no conflict of interest.

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
