# Peer review of "Outmigration Drives Cropland Decline and Woodland Increase in Rural Regions of Southwest China"

_land, doi:10.3390/land9110443_

Round 1

Reviewer 1 Report

My comments/suggestions attached. 

An Assessment of Outmigration-related Land Use Transition in Rural Area and Corresponding
Ecological Dynamics: A Case Study in Southwest China
This is a very interesting study combining commonly used methods (Mann-Kendall, Sen’s Slope and
Pearson’s Correlation) to understand the connection between rural outmigration and land use patterns
and resultant ecological dynamics.
Title
The title needs a minor edit. How about:
"An Assessment of Outmigration-related Land Use Transition in Rural Areas and Corresponding
Ecological Dynamics: A Case Study in Southwest China"
OR
"An Assessment of Outmigration-related Land Use Transition in Rural Southwest China and
Corresponding Ecological Dynamics"
Abstract
The abstract is well written: it introduces the research problem, states the objective of the study and
highlights major findings and conclusions. However, below are some suggested edits:
Line 13-14: “.... which has changed the regional labour force structure”.
Line 19: “.... a statistically significant increase in NDVI”.
Line22: “Only 2.5% of the area.......”
Line 23: I think there is an incorrect choice of word here. Do you mean "These results imply...."? Unless
you mean "This study concluded....". Besides, your statement is not very clear. Please consider reframing
your statement.
Line 26: I am not sure which declining trend you are referring to here. Please be specific in this statement.
Keywords
I would add "Land use and cover change", "Mann-Kendall", Sen's Slope" and "Linear regression"
Introduction
Authors clearly explained the background of the study and provided sufficient supporting facts and
numbers from published literature. They clearly state the objective of the study in the last paragraph of
the introduction. Below are some suggested edits
Line 40: Missing a word in "......China's uniqueness lies in that its agricultural....". Please revise.
Line 53: Consider changing to "....and have chosen to transfer their...".
Line 55-67: This paragraph is well written and supported by many interesting statistics!
Line 68-69:"........urban and closely adjacent areas have ....."
Line 78-81: I thought this statement is too long. Consider revising.
Line 83: "A better understanding of....".
Line 92-93: ".... south of China as a case study".
Study area, data and methodology
The study area is well described. Reasons why Rong County is the best case for this study are discussed.
One thing: you need to provide citations for most of the statements made in this section.
Line 128: "has decreased by 18.9% .....".
Very interesting graphs in Figure 2.
Data sources and images preprocessing
Line 145: "This study used data from..." or “This study utilized data from..."
Line 145-147: You cannot have a one-sentence paragraph. Please consider adding to this paragraph or
merging it with the next one.
Line 150: “All these senses” - it is not clear what you mean here. Please consider using another word here.
Line 155: "Choosing this period allowed the study to better....."
You do not mention the reason why you choose 2000, 2009 and 2018. Just add one line.
Line 159: You are mentioning NDVI and MODIS for the first time. It would be helpful to a new reader
that you write these in full and use the acronyms thereafter.
Line 160: You mention that MODIS NDVI has a temporal resolution of 16 days. MODIS data is usually
collected daily, but I believe what you used is a 16-day composite. Please consider revising this statement.
Line171-173: I believe these should be in the “Methods” section and not in the "Datasets and images
preprocessing”
Line 174-176: After moving the previous lines to methods, consider merging this with the remaining
segment.
Line 182: Consider providing a citation for Google Earth.
In section 2.3 you start using "we" but have said "the study" previously. Consider sticking to just one
connotation.
In section 2.3.2, I am suggesting that you merge the first and second paragraph since the second
paragraph is a continuation of the first.
Line 203: You are providing a reference here, but you have mentioned the linear regression model in line
201. Consider citing it on its first mention. Also consider providing a citation for the Mann-Kendall test as
mentioned in line 201.
I am suggesting that you merge the paragraphs starting on lines 209 and 214.
Line 232: ".....within the study area"
Line 239: "...can be beneficial for a deeper understanding......."
Line 247-254: it is not exactly clear what you are doing here. Please add more details of how this was
performed and why it was important.
Results
Line 266-268: please exclude these statements.
Line 298-300" ".... relatively more converted to ......". This part of the statement is not clear. Please revise.
In the results you provide a map of Z values from MK and a single value of Sen's Slope (I believe). I am
wondering if you performed Sen's Slope on the same NDVI raster time series like you did MK. Using
Sen's Slope on the raster time series will output a stack of five rasters, among them the slope and p-value
for each pixel. Here, you can mask out pixels with nonsignificant trends. Reading on, I see a slope map in
Figure 5. Is that Sen’s slope? A bit confusing, please clarify what you did.
Line 349: "Farmland" to "farmland".
Line 374-375: "..........and the improving living standard of local residents has further contributed to an
increase in the local residential area". It is not exactly clear what this statement means. What aspect of the
local residential area has increased? Is it the size, quality, or something else?
Line 433-434: ".... showed a statistically significant decreased trend of NDVI". Please rephrase it to
something like "..... showed a statistically significant decrease in NDVI".
Discussion
This study provides a very interesting discussion and also relates its results to results from other studies.
It also offers a comprehensive list of future research ideas to extend this study.
Line 526: "However, there still exist areas where improvements........"
Line 571: "Annual NDVI has been increasing at a speed of....." Consider using "rate" instead of "speed".
Line 577: "......were conducive for vegetation regeneration"
Conclusions
Conclusions made directly relate to results and discussion, and study implications were reemphasized
here.

Author Response

Thanks for your valuable comments. We have revised the relevant parts according to your suggestions. Please see the attachment.

Reviewer 2 Report

The rural outmigration is a global phenomenon, and it is closely linked to several social and demographic problems. The international significance of the topic is very important and should be better emphasized. Some examples of other countries (Mexico) should be mentioned. Authors need to better explain the international context and it is worth incorporating results from research on other continents. On the other hand, we get a detailed picture of the land conversion and the related processes in Southwest China. The reviewer confirms the authors’ goal and agrees with the main assumptions and findings. The paper is relatively well formulated. The authors present an accurate overview of the theme. The methods, the analyzes are thorough and well-illustrated. The discussion correlates to the starting points.

Author Response

Thanks for your comments. We have revised the relevant parts according to your suggestions. Please see the attachment.

Reviewer 3 Report

Although the authors consider that the fact that urban expansion tends to occupy a large portion of high-quality arable land (row 513) is paradoxical, this fact is an effect with "global" character in all territories where urbanization occurs in the rural spaces.

Another remarkable result is the conclusion that urbanization has caused a contraction both in the overall population and the proportions (380-381 rows and Figure 2) as well as the relationship between the land use transition process and emigration from the countryside to the city. But emigration itself has hardly been addressed. It is assumed but no detailed data is provided to support these claims.

The conclusion that urbanization led directly to migration from the countryside to the city is neither fully proven nor supported by data (row 565).

The authors do not adequately explain the correlation of cause and effect between the contraction of the general population and its consequences on the decrease of agricultural labor force and of extension of the lands for agricultural use.

Population dynamics and structural demographic changes should be quantified in more detail. Perhaps it would be appropriate to insist more on this type of effect, which is very different from that occurs in other areas, especially in Europe.

The authors state that their study has demonstrated the intrinsic links between land conversion, vegetation dynamics, and social factors (row 525), but in this study social factors have a much more limited treatment than those related to use of land and vegetation. The authors themselves seem to be aware that socioeconomic aspects, which they indicate as essential in the research objectives, have lower treatment in this paper and should be reinforced in subsequent research studies (row 533).

Author Response

(The authors gave the same response as above.)

Round 2

Reviewer 3 Report

The authors have introduced most of the suggested changes. I consider that the text has improved remarkably. I fully agree to the publication in present form.

Author Response

Thanks for your comments. We appreciate the valuable time you spent on our manuscript.